# A Comparative Study of Different Protocols for Isolation of Murine Neutrophils from Bone Marrow and Spleen

**DOI:** 10.3390/ijms242417273

**Published:** 2023-12-08

**Authors:** Khetam Sounbuli, Ludmila A. Alekseeva, Oleg V. Markov, Nadezhda L. Mironova

**Affiliations:** 1Institute of Chemical Biology and Fundamental Medicine SB RAS, Lavrentiev Ave. 8, 630090 Novosibirsk, Russia; khetam.sounbuli.edu@gmail.com (K.S.); alekseeva.mila.23@yandex.com (L.A.A.); markov_oleg@list.ru (O.V.M.); 2Faculty of Natural Sciences, Novosibirsk State University, 630090 Novosibirsk, Russia

**Keywords:** neutrophil isolation, Ficoll density gradient, immunomagnetic separation, positive selection, negative selection

## Abstract

Neutrophils are considered as the main player in innate immunity. In the last few years, it has been shown that they are involved in different physiological conditions and diseases. However, progress in the field of neutrophil biology is relatively slow due to existing difficulties in neutrophil isolation and maintenance in culture. Here we compare four protocols based on density-gradient and immunomagnetic methods for isolation of murine neutrophils from bone marrow and spleen. Neutrophil isolation was performed using Ficoll 1.077/1.119 g/mL density gradient, Ficoll 1.083/1.090/1.110 g/mL density gradient and immunomagnetic method of negative and positive selection. The different protocols were compared with respect to sample purity, cell viability, yield, and cost. The functionality of isolated neutrophils was checked by NETosis analysis and neutrophil oxidative burst test. Obtained data revealed that given purity/yield/viability/cost ratio the protocol based on cell centrifugation on Ficoll 1.077/1.119 g/mL density gradient is recommended for isolation of neutrophils from bone marrow, whereas immunomagnetic method of positive selection using Dynabeads is recommended for isolation of splenic neutrophils.

## 1. Introduction

Neutrophils are the first most abundant type of leukocyte in human blood and the second most abundant type in mice [1]. They represent the first line of defense against pathogens and are considered the main player in innate immunity [2]. Neutrophils have gained great interest in the last few years due to their novel roles in different physiological conditions and diseases [3,4]. Reconsideration of neutrophil homogeneity has been driven by the discovery of neutrophil extracellular traps (NETs). NETs are web-like structures containing neutrophilic DNA decorated with the content of granules and plays crucial role in neutrophil-mediated immune responses [5]. Moreover, the diversity of NET components (DNA, histones, lytic enzymes, antimicrobial peptides, and cytoskeleton proteins) ensures their participation in various physiological and pathological processes [6,7]. Moreover, NETs produced by tumor-associated neutrophils (TANs) in the tumor microenvironment was shown to be involved in cancer metastasis [8].

Despite its recently gained importance, progress in the field of neutrophil biology is relatively slow due to the difficulties accompanying neutrophil handling and maintenance in culture. Differentiated neutrophils have lost their capacity to proliferate and cannot be expanded in culture [9], and cannot be cryopreserved [10]. Moreover, they have limited ex vivo lifespan when maintained in culture [11]. In humans, a good source of neutrophils is peripheral blood. However, clinical material is not always available to researchers and therefore neutrophils are usually obtained from the bone marrow of mice. This is convenient due to the availability of the material and the absence of the need to obtain informed consent from the patient.

Several protocols for neutrophil isolation from murine blood or bone marrow have been described [12,13,14,15]. Classical protocols are based on a density-gradient centrifugation, where cell suspensions or blood are centrifuged on a density gradient and cell moves to a layer with a density equal to their own. The method of immunomagnetic cell separation was also used; it consists in tagging the cell of interest (positive selection) or contaminating cells (negative selection) with magnetic beads and their separation in a magnetic field [16]. Selecting the right protocol depends on various parameters such as sample purity, budget, and timing. Moreover, an important parameter to consider is the functionality of isolated cells. Where the classical density-gradient centrifugation may affect neutrophil functionality, immunomagnetic methods of negative selection are believed to gain quiescent and native cells with higher sample purity [10,17]. Thus, the isolation of viable neutrophils with good functional characteristics is a challenging task.

In this study, we compare four protocols including density-gradient centrifugation and immunomagnetic methods for neutrophil isolation from mouse bone marrow and spleen. The bone marrow was used as neutrophil source because it serves the store for a high quantity of quiescent ones. Also, the spleen was chosen as a source of neutrophils, since it is the second largest source of neutrophils after the bone marrow and one of the primary sites of neutrophil clearance along with bone marrow and liver [18,19]. Moreover, unlike bone marrow-derived neutrophils, which are believed to be naive, splenic neutrophils could reflect the real profile of neutrophils in organisms with different pathologies. In addition to neutrophils undergoing clearance in the spleen, spleen contains spleen-resident neutrophils which were suggested to participate in tissue regulatory functions under homeostasis and according to speculation serve as reservoirs of granulocyte-like myeloid-derived suppressor cells (G-MDSC) and/or tumor-associated neutrophils (TANs) under cancer settings [18,20,21]. In this study, we compare the yield, sample purity, and viability of bone marrow- or spleen-derived neutrophils isolated using different protocols and evaluate their costs.

## 2. Results

### 2.1. Protocol Selection for Neutrophil Isolation

For neutrophil isolation, multiple protocols were applied: (1) protocols using density gradient centrifugation, (2) and immunomagnetic methods of positive and negative selection. Density gradient centrifugation Protocols taken from [12,13,22] were slightly modified. We analyzed and compared the protocols of neutrophil isolation from C57Bl/6 mice and used bone marrow and spleen as the source of neutrophils. The used protocols applied for neutrophil isolation are presented in Figure 1.

Modifications included maintaining a constant temperature regimen and performing all procedures at room temperature to avoid neutrophil activation [14] and under sterile conditions to avoid or diminish contamination. Various solutions, Ficoll, Percoll, Histopaque, Optiprep, and Lympholyte, could be used in density-gradient centrifugation protocols. We selected Ficoll as the simplest, most accessible, and the most widely used medium for density-gradient centrifugation [23]. Classical density-gradient purification of neutrophils was performed using either Ficoll 1.077/1.119 g/mL based two-layer solution (2FLG protocol) or Ficoll 1.083/1.090/1.110 g/mL based three-layer solution (3FLG protocol) (Figure 1B). As mentioned in the Methods section, the centrifugations were performed at room temperature, without acceleration and brake, to avoid layer mixing, to achieve better sedimentation of the more floatable neutrophil fractions, and to avoid layer erosion.

Among immunomagnetic methods, we compared neutrophil isolation using negative neutrophil selection using the EasySepTM Mouse Neutrophil Enrichment Kit (INS protocol) and positive neutrophil selection using Dynabeads (IPS protocol) (Figure 1C).

### 2.2. Characterization of the Samples Isolated from Bone Marrow and Spleen

Purity of isolated cells in the samples were assessed using flow cytometry. Neutrophils were defined as CD11b and Ly6G double positive events.

BMCs were prepared from the bone marrow of C57BL/6 mice, and neutrophils were isolated using 2FLG/3FLG and immunomagnetic protocols. The flow cytometric characterization of the obtained neutrophil samples is presented in Figure 2. The Ficoll protocols resulted in close purities: 72.4 ± 1.7% for 2FLG protocol (Figure 2A), and 76.7 ± 5.0% for 3FLG protocol (Figure 2B). The INS method also resulted in high-purity neutrophil population reached 80.3 ± 0.3% (Figure 2C). As can be seen, the highest sample purity was obtained using IPS method as purity was 99.3 ± 0.3% (Figure 2D).

Splenocytes were prepared from the spleens of C57BL/6 mice, and neutrophils were isolated using the same four protocols as for BMCs. The characteristics of neutrophils isolated from the spleen are presented in Figure 3. It should be noted that the samples isolated from the spleen were characterized by a lower purity compared with the samples isolated from the bone marrow. The Ficoll-based protocols were unsuitable for splenic neutrophil isolation with sample purity about 10% (Figure 3A,B). The INS protocol resulted in a sample purity of 48.6 ± 4.3% (Figure 3C). The IPS protocol was the most effective, and the isolated sample had a purity of 98.7 ± 0.9% (Figure 3D).

### 2.3. Comparison of Purity, Viability, and Yield of Neutrophil Samples Isolated from Bone Marrow and Spleen Using Different Protocols

The protocols applied for neutrophil isolation were compared with respect to sample purity, cell viability, and the yield of neutrophils, as well as the cost (Table 1 and Table 2).

The 2FLG and 3FLG protocols resulted in similar sample purity of approximately 75%. The application of the INS protocol results in samples with a higher sample purity 80.3 ± 0.3% (Table 1). However, the highest sample purity (99.3 ± 0.3%) was achieved using the IPS protocol. None of the protocols affected cell viability, which was approximately 90% in all samples. In Table 1, general and neutrophil yields are provided. The general yield is the cell yield obtained after neutrophil isolation without considering sample purity. Neutrophil yield was calculated from the general yield based on purity. As seen from the Table 1 neutrophil yield increase in order 3FLG << IPS < 2FLG < INS, and the sharper difference (more than 14 fold) in neutrophil yield was between 3FLG and 2FLG protocols showing the applicability of 2FLG protocol for neutrophil isolation from bone marrow. The only disadvantage of INS protocol is its high cost.

Similar analysis was performed for neutrophil isolation from spleen (Table 2).

The obtained data revealed that both density gradient protocols allowed us to obtain samples with extremely low purity (about 10% of splenocytes). It should be noted that the INS protocol, which demonstrates high purity in neutrophil isolation from bone marrow, when applied for neutrophil isolation from the spleen, gives neutrophil samples with a purity of only 50% (Table 2). This was an expected result because the INS protocol is not optimized for the spleen. Cell viability slightly varied between the protocols from 74% to 89% but was lower than in BM-derived samples. Neutrophil yield was also extremely low for all protocols used (Table 2).

Optimization of the INS protocol for splenic neutrophil isolation could be applied to obtain untouched splenic neutrophils; however, it could be rather expensive because of the complex cocktail of antibodies that should be added to the splenocyte suspension along with the EasySep antibody cocktail [24]. Taking into account the low content of neutrophils in the spleen of healthy mice [24,25,26] and obtained data on purity/viability of cells (Table 2) the IPS protocol is recommended for splenic neutrophil isolation.

### 2.4. Ability of BM-Derived Neutrophils Isolated by the 2FLG Protocol and Splenic Neutrophils Isolated Using the IPS Protocol to Produce NETs

One important functional test for neutrophils is the ability to form NETs. Density gradient protocols unlike immunomagnetic methods of negative selection could impair the functionality of isolated neutrophils, which prompted us to study the ability of neutrophils isolated by 2FLG protocol to produce NETs.

BM-derived neutrophils isolated by 2FLG protocol were treated with 100 μg/mL LPS, 50 or 500 nM PMA, the activator of protein kinase C [27], or 5 μM Ca^2+^ ionophore A23187 for 3 h.

The unstimulated neutrophils showed preserved morphology (Figure 4A). The obtained data revealed that neutrophils are capable of NET release under the action of physiological activator LPS (Figure 4B) and chemical activators: PMA (Figure 4C,D) and A23187 (Figure 4E). LPS at a high concentration (100 μg/mL) seemed to be the less effective in inducing NETosis (Figure 4B). Neutrophil treatment with 50 nM PMA led to the formation of filament NETs (Figure 4C). Neutrophil treatment with 500 nM PMA lead to significant morphological changes in neutrophils, cell enlargement, and bubbling. In the case of 500 nM PMA, along with filament NET formation, nuclei enlargement and chromatin diffusion were detected (Figure 4D), which can be described as “diffused” NETs or chromatin “cloud”, and could be an early sign of NETosis [28,29,30] (Figure 4D). Filament-like NETs were also detected in A23187-activated samples (Figure 4E). BM-derived neutrophils isolated using the INS protocol responded in a similar manner to BM-neutrophils isolated using the 2FLG protocol (primary).

Positive immunomagnetic selection methods could affect the functionality of isolated cells because magnetic particles are linked to a particular cell membrane protein, which, of course, plays a role in cell function. In our study, splenic neutrophils were isolated on Dynabeads, which were linked to Ly6G. Ly6G is known to participate in neutrophil adhesion and immune responses against pathogens [31]. Microscopic analysis revealed some alterations in neutrophil morphology in the control samples, which confirmed the fact that neutrophils are slightly activated (Figure 5A). Treatment of spleen-derived neutrophils with LPS, PMA, or A23187 resulted in the appearance of cell changes in a manner similar to that of BM-derived neutrophils isolated using the density-gradient protocol.

These findings were further confirmed by counting the percentage of neutrophils producing NETs in the form of extracellular filament-like DNA (Figure 6). Despite the source of neutrophils or the isolation protocol, there was no significant increase in NET formation under the effect of LPS compared with the control (Figure 6). BM-marrow-derived neutrophils isolated using the 2FLG or the INS protocols responded in a similar manner to chemical stimulants; however, neutrophils isolated by the INS protocol more effectively formed NETs compared with neutrophils isolated by the 2FLG methods: 82.9 ± 5.9% of neutrophils formed NETs in the case of INS vs. 46.8 ± 11.2% in the case of 2FLG in response to A23187, whereas the percentage was 47.6 ± 4.7% vs. 23.3 ± 3.3, respectively, in response to 50 nM PMA. No significant increase was observed in response to 500 nM PMA (Figure 6).

LPS also did not stimulate NETosis in splenic neutrophils (Figure 6C). PMA at high concentration and A23187 caused a significant increase in the percentage of splenic neutrophils forming NETs: 24.2 ± 6.1% and 38.7 ± 5.6%, respectively, in comparison with 9.7 ± 1.9% in the control.

### 2.5. ROS Production in BM-Derived Neutrophils Isolated Using the 2FLG Protocol and Splenic Neutrophils Isolated Using Dynabeads in Response to Stimulation

We assessed the ability of isolated neutrophils to produce ROS in response to A23187 using a DCFDA-based ROS detection assay. Within 90 min, ROS accumulated at a higher level in stimulated neutrophils than in untreated neutrophils, isolated from the bone marrow or spleen. Moreover, ROS baseline levels in activated neutrophils were higher than those in controls, and further ROS accumulation occurred at a faster rate. It was also found that the ROS baseline level in splenic neutrophils is 2–2.5 times higher than that in bone marrow neutrophils (Figure 7), which may reflect the maturation state of splenic neutrophils.

## 3. Discussion

Despite advances in the field of neutrophil biology, various questions on neutrophil handling are exist. In our work, we focused on optimizing murine neutrophil isolation protocols to achieve precise procedures that ensure high yield and viability and used mouse bone marrow and spleen as neutrophil sources. Due to the low percentage of neutrophils in mouse peripheral blood, approximately 10–30%, and the low yield [12,32,33], we did not use this source in our study.

Bone marrow is considered as a store of quiescent nonpolarized neutrophils. Isolated neutrophils from the bone marrow are a good choice for studying neutrophil biology and function. When analyzing the protocols for neutrophil isolation, we noticed that several modifications can affect the isolated population. For example, Heib et al. found that bone marrow flushing or centrifugation could lead to different cell yields [34]. In addition to the fact that different protocols lead to different sample purity, yield, and viability, they can affect the function and phenotype of isolated cells [10]. Classical density gradient isolation protocols are widely used to isolate bone marrow-derived neutrophils and provide sufficient purity [13]. As a result of our study, we can suggest protocol of gradient centrifugation on two-layer Ficoll 1.077/1.119 density gradient for neutrophil isolation from bone marrow which demonstrated high purity, yield and cell viability and is considered as budget choice.

It should be noted that the cells isolated by immunomagnetic method of negative selection are untouched and believed to be quiescent and native cells, whereas density gradient protocols could affect the functionality of isolated neutrophils [10,17], which led us to investigate the ability of isolated neutrophils using Ficoll 1.077/1.119 to produce NETs. The neutrophils isolated using this protocol responded to activation with PMA and A23187, indicating their preserved functionality; however, they were less efficient in comparison with immunomagnetic methods of negative selection (Figure 6A,B). In response to a low concentration of PMA (50 nM), neutrophils produced NETs with comet-like shape and filaments. At higher concentrations (500 nM), neutrophils enlarged and showed signs of plasma membrane permeabilization. This could be explained by the shedding of plasma membrane microvesicles upon neutrophil activation, a process which may be regulated by µ-calpain and ezrin proteins during NETosis [35,36,37]. Moreover, the nuclei were enlarged and the genetic material was decondensed, which could be described as “diffused” or chromatin “cloud” NETs [28,29]. However, filament-like NETs were approximately absences in the samples treated with 500 nM PMA. PMA is a potent activator of neutrophils and acts by activating protein kinase C (PKC) [27]. PMA is believed to induce suicidal NETosis, as first described by Takei et al. [38]. However, the actions of PMA are believed to be diverse and may differ according to the isoforms of PKC [39]. Our results showed that neutrophil responses to PMA may differ with respect to PMA concentration. The lower efficacy of PMA at high concentrations in inducing NETosis may be explained by the fact that high concentrations of PMA could cause trauma in neutrophils, and they could not function normally. LPS, a gram-negative bacterial stimulus known to activate vital NETosis [40], did not induce NETosis in our study. However, the ability of LPS to induce NET production is contradictory [41]. These inconsistent findings could be explained by the structural diversity of LPS from different bacterial strains, which are known to induce heterogeneous immune responses [29,42]. In our study, LPS from Escherichia coli serotype O55:B5 was used. This serotype did not stimulate NETosis in the study of Pieterse et al. [29], which is consistent with our finding, although in their study the authors investigated human neutrophils, whereas we used murine. The most potent activator was A23187, which acts by increasing Ca^2+^ concentration in the cytosol [43]. Although A23187 is believed to induce NOX-independent NETosis, neutrophils produce ROS in response to A23187 activation [44] (Figure 7).

Splenic neutrophils are of great interest. Unlike naïve bone marrow-derived neutrophils, splenic neutrophils are mature, and because of their clearance to the spleen after performing their function in the organism, splenic neutrophils could reflect the real profile and polarization status of neutrophils in different pathologies. Splenic neutrophils comprise two neutrophil populations: spleen-resident neutrophils and a population of neutrophils originated from the process of neutrophil clearance into the spleen. In healthy mice, splenic neutrophils count for <10% of all splenocytes, a percentage, which goes higher under different diseases setting [24,25,26,45]. Splenic neutrophils are considered an important player in regulating immune responses [46]. Moreover, they could represent TANs or MDSCs in some speculations [21]. In our studies, as well as in the studies of other authors [24], it was shown that classical protocols do not allow isolating a pure population of neutrophils from the spleen. Since the classical protocols were insufficient to gain a pure neutrophil population, different techniques were studied. The first suggestion is using fluorescence-activated cell sorting (FACS). However, FACS is a time-consuming process, which makes it impractical, especially when analyzing large cohorts [47]. Here, we tried to use the immunomagnetic positive selection to isolate splenic neutrophils. The procedure is fast and results in a high-purity sample. Cells isolated using immunomagnetic positive selection represent a good source for transcriptional and functional analyses.

Splenic neutrophils isolated using the immunomagnetic method of positive selection turn out to be slightly activated. This is likely because the isolation method which involves antibodies connecting magnetic beads to the neutrophils. Nevertheless, splenic neutrophils respond to stimuli and form NET structures in a manner similar to BM-derived neutrophils isolated using the density-gradient protocol.

However, when studying neutrophil functions in culture, it is better to use untouched neutrophils. The commercial kits based on the immunomagnetic negative selection method failed to isolate a pure neutrophil population most likely because of the higher frequency of contaminating cell populations and lower frequency of neutrophils in the spleen compared with bone marrow or blood, for which the kits are specified. In the work of Coquery et al. [24], the authors modified the panel of antibodies used for negative selection of splenic neutrophils. They added anti-CD3, anti-CD19, and anti-NK1.1 antibodies to the antibody isolation cocktail and increased the concentrations of anti-Ter119, anti-B220, anti-F4/80 and anti-CD11c antibodies in the antibody isolation cocktail [24]. These modifications allowed to obtain high sample purity; however, the high cost of such an approach is a significant limitation.

Analysis of neutrophil functions included two functional tests of neutrophils: NETosis and ROS production analysis, which are most commonly used to study neutrophil function. The functional tests of neutrophils include phagocytosis, microbe killing, degranulation assays, and chemotaxis. Blanter et al. [10] compared the functionality of human neutrophils, isolated on Ficoll or by immunomagnetic separation, using different tests and found that immunomagnetic separation of neutrophils is more suitable for studying neutrophil polarization, phagocytosis, ROS production, degranulation, and NETosis, whereas density gradient purification is preferred for Boyden chemotaxis assays. Our results showed that in the case of NETosis or oxidative burst analysis, the protocols recommended in this study could be applied. However, when analyzing other neutrophil functions, there is no doubt that immunomagnetic methods of negative selection may be more suitable because they are faster, easier, and produce untouched neutrophils.

In conclusion, in this study we described in detail different protocols based on density gradient centrifugation and immunomagnetic cell separation for isolation of murine neutrophil from bone marrow and spleen. Obtained results allow researchers in this field to choose the optimal protocol for different goals based on sample purity, cell functionality, cost and subsequent purposes for which neutrophils were obtained.

## 4. Materials and Methods

### 4.1. Mice

C57Bl/6 male mice aged 3–4 months were obtained from the vivarium of ICBFM SB RAS (Novosibirsk, Russia). Mice were housed in plastic cages under normal daylight conditions. Water and food were provided ad libitum. All animal procedures were carried out in strict accordance with the recommendations for proper use and care of laboratory animals (ECC Directive 2010/63/EU). The experimental protocols were approved by the Committee on the Ethics of Animal Experiments with the Institute of Cytology and Genetics SB RAS (ethical approval number 49 from 23 May 2019), and all efforts were made to minimize suffering.

### 4.2. Bone Marrow Cell (BMC) Isolation

The protocol was adopted from Swamydas et al. [13] and Ubags et al. [14] with some modifications. All procedures were performed at room temperature, and all solutions used were pre-equilibrated at room temperature [14]. After mouse euthanasia, the mouse was placed in a supine position and the skin was treated with ethanol 70%. An incision was made in the skin around the tibias, and the skin from the mouse leg was removed. After removing the skin around the leg, the femur was disconnected from the hip joint. The femur was disconnected from the tibia, and the muscles around the bones were removed. The bones were placed in 70% ethanol in a Petri dish for a few seconds and then washed in sterile PBS (Servicebio, Wuhan, China) in a Petri dish. The epiphyses were cut, and BMCs were flushed using an insulin U-100 syringe filled with RPMI 1640 (ThermoFisher Scientific, Waltham, MA, USA) supplemented with 10% FBS (BioFroxx, Einhausen, Germany), 1% antibiotic-antimycotic solution ((penicillin (10,000 IU/mL), streptomycin (10 mg/mL), and amphotericin B (25 μg/mL)) (MP Biomedicals, Santa Ana, CA, USA), and 2 mM EDTA (MP Biomedicals, Santa Ana, CA, USA) (RPMI/FBS/EDTA). BMCs were resuspended and centrifuged at 1400 rpm for 7 min. For red blood cell lysis, the cell pellet was resuspended in 1.5 mL lysis buffer containing 0.15 M NH_4_Cl, 10 mM NaHCO_3_, and 0.1 mM EDTA (Sigma-Aldrich, Darmstadt, Germany) and incubated for 7 min at room temperature. PBS was added up to 15 mL to neutralize the lysis buffer and the cell suspension was centrifuged at 1400 rpm for 7 min. The cells were washed twice with RPMI/FBS/EDTA solution and resuspended in the required buffer according to the following protocol.

### 4.3. Splenocyte Suspension Preparation

The euthanized mouse was placed in a supine position, and the skin was treated with ethanol 70%. An incision was made in the skin around the midline of the mouse on the left side. The spleen was harvested and placed in a sterile Petri dish containing 1 mL of RPMI/FBS/EDTA. Spleen homogenization was performed mechanically using the thumb rest side of a syringe’s plunger. After the spleen was fully dissociated, the cell suspension was filtered through a 70 μm cell strainer (Corning, Glendale, AZ, USA), and red blood cell lysis was performed as described above.

### 4.4. Preparation of Ficoll Solutions of Different Densities

Ficoll solutions with densities of 1.077, 1.083, 1.090, 1.110, and 1.119 g/mL were prepared by dissolving Ficoll 400 powder (Sigma-Aldrich, Darmstadt, Germany) in PBS at room temperature. The density of the solutions was measured using a hydrometer (Steklopribor, Zavodskoe, Ukraine).

### 4.5. Neutrophil Isolation

#### 4.5.1. Ficoll 1.077/1.119 g/mL Density Gradient (Two Layer Density Gradient Protocol, 2FLG Protocol)

Suspensions of BMCs or spleenocytes were applied onto two-layer Ficoll gradient (1.077 and 1.119 g/mL) and centrifuged for 30 min at 2000 rpm at 25 °C. Neutrophils were collected at the interface between Ficoll 1.077 and Ficoll 1.119 and washed twice with RPMI/FBS/EDTA solution [13].

#### 4.5.2. Ficoll 1.083/1.090/1.110 g/mL Density Gradient (Three Layer Density Gradient Protocol, 3FLG Protocol)

The protocol was adopted from Boxio et al. [22] and Marchi et al. [12], where Percoll was replaced by Ficoll. The cell pellet obtained as described above was resuspended in PBS supplemented with 1% BSA (HyClone, Washington, WA, USA) and 15 mM EDTA and was applied on a three-layer Ficoll gradient (1.083, 1.090 and 1.110 g/mL) and centrifuged at 1500× *g* for 30 min at room temperature. Neutrophils from the 1.090/1.110 interface and the upper part of the 1.110 layer were harvested after carefully removing the cells from the upper phases. Neutrophils were washed twice with PBS supplemented with 1% BSA and 15 mM EDTA.

#### 4.5.3. Immunomagnetic Methods

##### Immunomagnetic Negative Selection (INS Protocol)

Neutrophils from BMC or splenocyte suspensions were isolated using the EasySepTM Mouse Neutrophil Enrichment Kit (StemCell Technologies, Vancouver, BC, Canada) according to the manufacturer’s instructions with some modifications. Briefly, after red blood cell lysis, the cells were resuspended in PBS supplemented with 2% FBS and 1 mM EDTA (PBS/FBS/EDTA) at a concentration of 1 × 10^8^ cells/mL. Rat serum and enrichment cocktail were added to the sample (50 μL/mL of sample) and the sample was incubated for 15 min at 4 °C. The cells were washed in PBS/FBS/EDTA solution and centrifuged at 300× *g* for 10 min. The cell pellet was resuspended in the original volume of PBS/FBS/EDTA solution, the biotin selection cocktail was added to the sample (50 μL/mL of sample), and the sample was incubated for 15 min at 4 °C. Magnetic particles were resuspended, added to the sample (150 μL/mL of sample), and the sample was incubated for 10 min at 4 °C. After incubation, the sample was topped up to 2.5 mL of PBS/FBS/EDTA solution, mixed by pipetting, and the tube was placed into the magnet (ThermoFisher Scientific, Waltham, MA, USA) and incubated for 3 min at room temperature. Neutrophil suspension was placed into a new tube.

##### Immunomagnetic Positive Selection (IPS Protocol)

Neutrophils from the bone marrow or spleen were isolated using Dynabeads Sheep anti-Rat IgG (ThermoFisher Scientific, Waltham, MA, USA) according to the manufacturer’s instructions with some modifications. In brief, the cell pellets obtained as described above were resuspended in 500 μL PBS supplemented with 2% FBS and 1 mM EDTA. A total of 1 μg or 2 μg of rat IgG anti-mouse Ly6G (PE, Sony Biotechnology, Tokyo, Japan, Cat# 1238035) was added to the spleen or bone marrow suspension, respectively, and the sample was incubated for 20 min at 4 °C. After labeling, the cells were washed and resuspended in 1 mL of PBS supplemented with 0.1% BSA and 2 mM EDTA (PBS/BSA/EDTA). A total of 10 μL or 20 μL of Dynabeads Sheep anti-Rat IgG were added to the labeled cells of spleen or bone marrow, respectively, and the cells were incubated for 20 min at 4 °C. After incubation, the volume was doubled by adding 1 mL of PBS/BSA/EDTA and the tube was placed in the magnet (ThermoFisher Scientific, Waltham, MA, USA) for 2 min. The supernatant was discarded, and the bead-bound cells were washed 4 times by adding 1 mL of PBS/BSA/EDTA, placing the tube in the magnet for 1 min and discarding the supernatant.

### 4.6. Neutrophil Characterization

#### 4.6.1. Sample Purity

The purity of neutrophil samples was assessed by flow cytometry analysis. The samples were incubated with anti-CD11b (PerCP/Cy5.5, BD Biosciences, San Jose, CA, USA, Cat#550993) and anti-Ly6G (PE, Sony Biotechnology, Tokyo, Japan, Cat#1238035) antibodies and analyzed using a NovoCyte 3000 flow cytometer (ACEA Biosciences, San Diego, CA, USA). Data were processed using NovoExpress software v. 1.1.0 (ACEA Biosciences, San Diego, CA, USA). Singlets were selected from the FSC-H versus FSC-A dot plot in the debris exclusion gate (gate 1), and in singlet gate CD11b (PerCP-Cy5.5) and Ly6G (PE) double positive events were gated to detect neutrophils.

#### 4.6.2. Cell Viability

Cell viability was assessed using trypan blue exclusion assay (Shanghai Macklin Biochemical Technology Co., Ltd., Shanghai, China). For samples isolated using IPS protocol, yield and viability were measured in a Goryaev chamber using an Axiostar plus microscope (Zeiss, Munich, Germany). Yield and viability of neutrophils in the other samples were measured using TC20 automated cell counter (Bio-Rad, Hercules, CA, USA).

### 4.7. Neutrophil Extracellular Trap Visualization

For NET formation, neutrophils isolated from BMC samples on a Ficoll 1.077/1.119 g/mL density gradient or using EasySepTM Mouse Neutrophil Enrichment Kit (StemCell Technologies, Vancouver, Canada) or splenic neutrophils isolated using Dynabeads were used. After isolation, cells were washed twice with RPMI 1640 supplemented with 5 mM HEPES (Sigma-Aldrich, Darmstadt, Germany), 1 mM sodium pyruvate (ThermoFisher Scientific, Waltham, MA, USA), and 1% antibiotic-antimycotic solution.

Sterilized glass coverslips were placed into the wells of 24-well plate and treated with poly-L-lysine solution (Sigma-Aldrich, Darmstadt, Germany) for 20 min. Poly-l-lysine solution was removed and the coverslips were washed with PBS. A total of 100,000 neutrophils were suspended in 500 μL RPMI supplemented with 1% antibiotic-antimycotic solution, 5 mM HEPES, and 1 mM sodium pyruvate. Neutrophils were supplemented with 50 or 500 nM PMA, 100 μg/mL LPS (Sigma-Aldrich, Darmstadt, Germany) or 5 μM ionophore A23187 (Abcam, Cambridge, UK), placed on the coverslips in 24-well plates, and incubated at 37 °C in a humidified atmosphere with 5% CO_2_ for 3 h. Then 100 μL of 24% formaldehyde (Sigma-Aldrich, Darmstadt, Germany) was added to the wells and incubated for 30 min at room temperature for sample fixation. After fixation, the medium was removed, washed with PBS, and 0.1% TritonX100 were add and the samples were incubated for 2 min at room temperature for cell permeabilization. After permeabilization, blocking was performed to reduce unspecific reactions by incubating in blocking buffer (1% FBS, 1% BSA in PBS [48]) for 30 min at room temperature followed by FcR III and FcR II blocking with Rat anti-mouse anti-CD16/32 antibodies at 1:200 dilution (Elabscience, Houston, Texas, USA, Cat#E-AB-F0997A) for 30 min at room temperature. After blocking, samples were incubated with Rabbit anti-mouse anti-myeloperoxidase antibodies at 1:50 dilution (Abcam, Cambridge, UK, Cat#ab208670) for 1 h at 37 °C. The coverslips were washed with PBS twice and stained with secondary antibodies Goat Anti-Rabbit IgG Alexa Fluor^®^ 680 at 1:500 dilution for 1 h at 37 °C (Abcam, Cambridge, UK, Cat#ab175773). After, the coverslips were washed with PBS twice and treated with a lipophilic dye DIOC6 solution (0.6 μg/mL) (Abcam, Cambridge, UK) and the cells were incubated at 37 °C in the dark for 20 min. After incubation, the supernatant was discarded and the coverslips were washed in PBS, removed from the well, and flipped on a 10 μL of DAPI Fluoromount-G^®^ (SouthernBiotech, Birmingham, AL, USA) placed on a glass slide. The slides were placed in the dark at room temperature in horizontal position for one night and then analyzed by confocal fluorescence microscopy LSM710 (Zeiss, Munich, Germany) using a plan-apochromat 63×/1.40 Oil DIC M27 objective. The obtained images were analyzed using ZEN software 2012 (Zeiss, Munich, Germany) and ImageJ software version 1.54d (Wayne Rasband and contributors, NIH, Madison, WI, USA).

For NET quantification, the images were obtained with confocal fluorescent microscopy LSM710 (Zeiss, Munich, Germany) using an EC Plan-Neofluar 20×/0.50 M27 objective, and cells in at least 5 non-overlapping fields were counted for each condition. The results are expressed as the percentage of neutrophils forming NETs. NETs were identified as extended DNA filaments released from neutrophils or in decondensed cloud-like extracellular DNA.

### 4.8. Neutrophil Oxidative Burst Test

To evaluate ROS production in neutrophils 2′,7′-dichlorofluorescin diacetate (DCFDA) (Sigma-Aldrich, Darmstadt, Germany) was used. DCFDA penetrates the cell membrane and is then deacetylated by intracellular esterases to a non-fluorescent compound. This compound is oxidized by ROS to form 2′,7′-dichlorofluorescein (DCF), a highly green florescent product [49]. BM-derived neutrophils isolated using the 2FLG protocol and splenic neutrophils isolated using the IPS method were incubated with 10 μM DCFDA for 30 min at 37 °C. After DCFDA preloading, cells were washed with PBS and either left unstimulated or treated with 5 μM A23187 (ab120287, Abcam, Cambridge, UK) and immediately transferred to a 96-well plate at a density of 10,000 cells per well. Fluorescence was measured at 10-min intervals up to 90 min using a CLARIOstar Plus plate reader (BMG LABTECH, Ortenberg, Germany).

### 4.9. Statistical Analysis

All experiments were performed in triplicate. Statistical analysis was performed using GraphPad Prism version 7.00 (GraphPad Software, version 7.00, San Diego, CA, USA). Neutrophil population characteristics in Section 2.2 and Section 2.3 are presented as mean ± SEM. NET quantification results are presented as mean ± SEM and were analyzed using Tukey’s multiple comparisons test. Florescence intensity values are shown as mean ± SD.

## Figures and Tables

**Figure 1 ijms-24-17273-f001:**
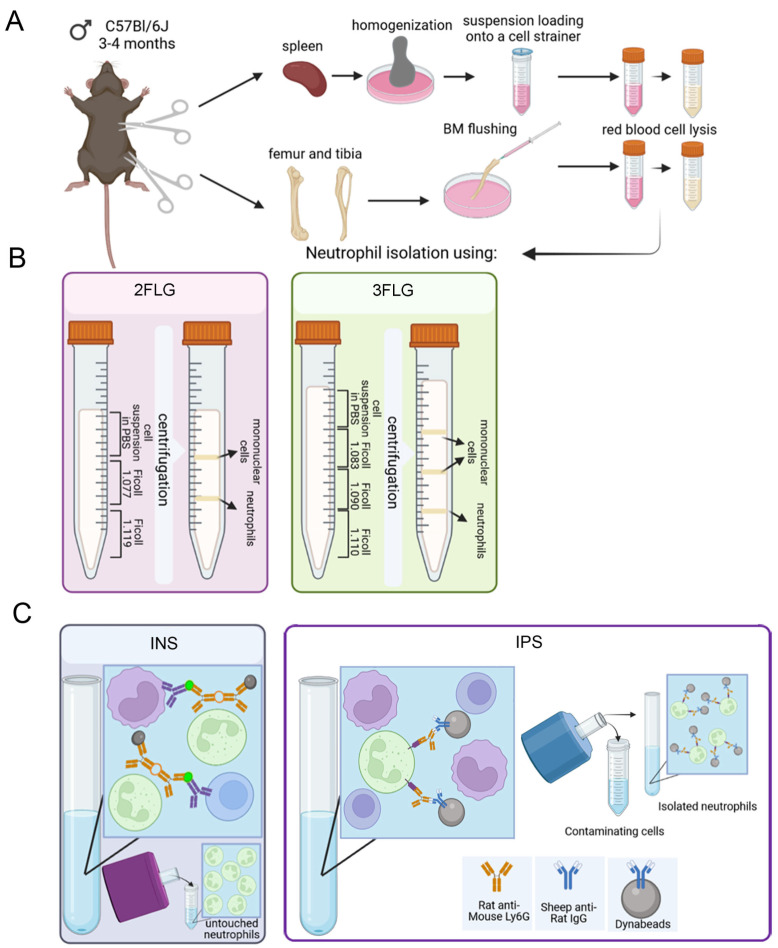
Scheme showing the different protocols for neutrophil isolation from the bone marrow and spleen of mice used in this study. (**A**) Bone marrow and spleen cell suspension preparation. (**B**) Ficoll density gradient protocols. (**C**) Immunomagnetic protocols.

**Figure 2 ijms-24-17273-f002:**
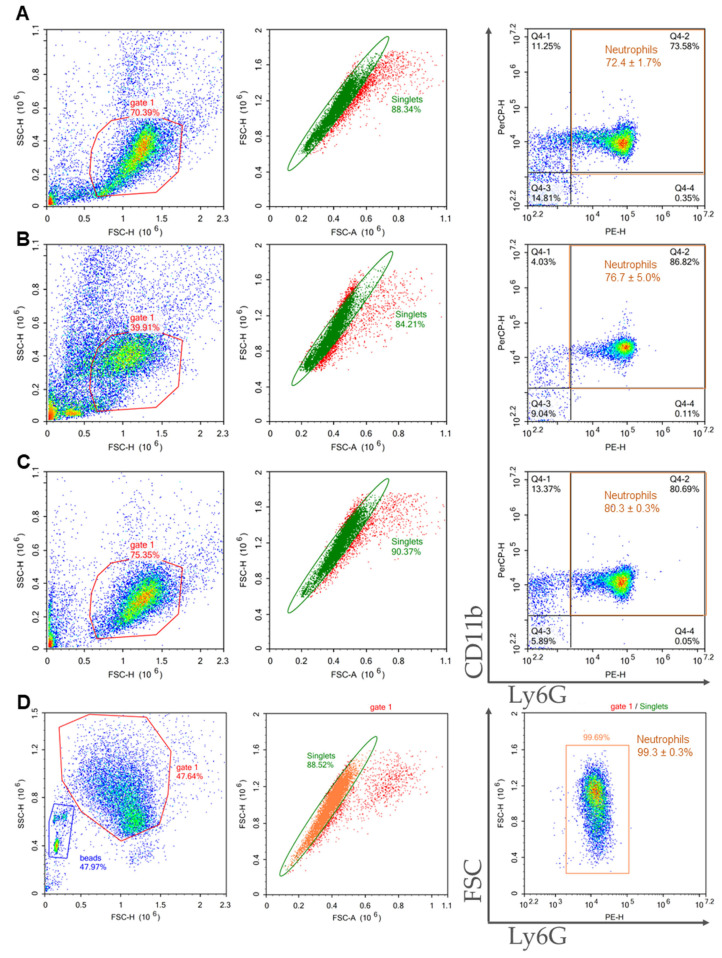
Representative dot plots of the flow cytometry gating strategy of bone marrow-derived neutrophils isolated using 2FLG (**A**), 3FLG (**B**), INS (**C**) and IPS protocols (**D**). Debris was excluded. In the debris exclusion gate (gate 1), singlets were selected from the FSC-H versus FSC-A dot plot. In singlet gate neutrophils were detected as double positive events for CD11b (PerCP-Cy5.5) and Ly6G (PE).

**Figure 3 ijms-24-17273-f003:**
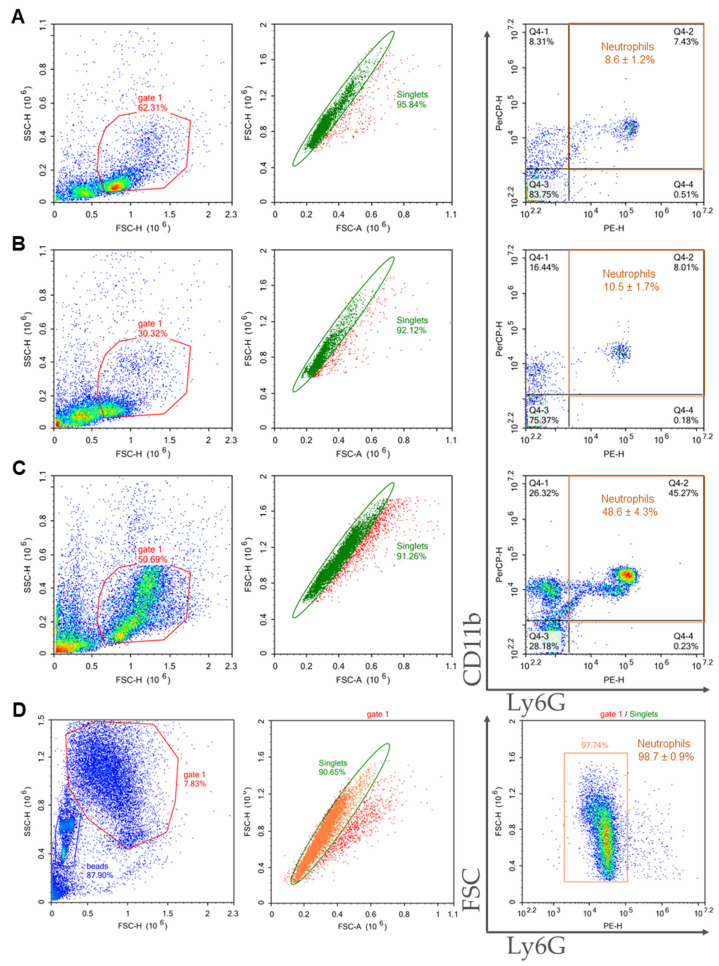
Representative dot plots of the flow cytometry gating strategy of spleen-derived neutrophils isolated using 2FLG (**A**), 3FLG (**B**), INS (**C**), and IPS (**D**) protocols. Debris was excluded. In the debris exclusion gate (gate 1), singlets were selected from the FSC-H versus FSC-A dot plot. In singlet gate neutrophils were detected as double positive events for CD11b (events PerCP-Cy5.5) and Ly6G (PE).

**Figure 4 ijms-24-17273-f004:**
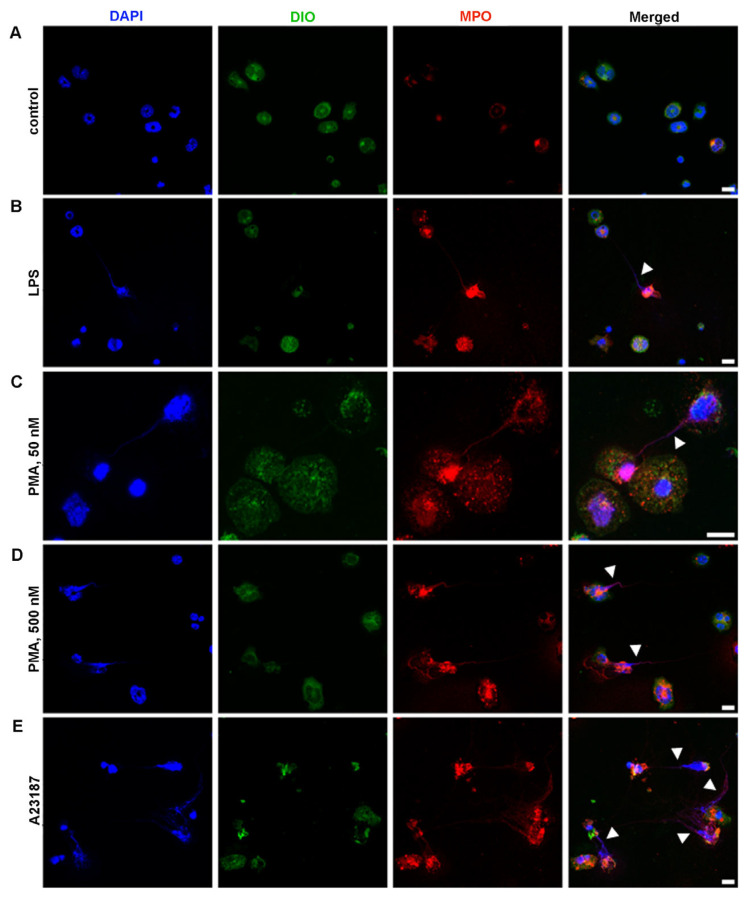
NET formation by BM-derived neutrophils, isolated by 2FLG protocol. (**A**) Unstimulated neutrophils. (**B**) Neutrophils stimulated with 100 μg/mL LPS. (**C**,**D**) Neutrophils stimulated with 50 nM and 500 nM PMA, respectively. (**E**) Neutrophils stimulated with 5 μM A23187. Neutrophils were labeled with DAPI (blue signal, nuclei), DIOC6 (green signal, membranes) and anti-MPO (Red signal). Scale bar, 10 µm. Arrowheads indicate NETs. Images were obtained by confocal fluorescent microscopy.

**Figure 5 ijms-24-17273-f005:**
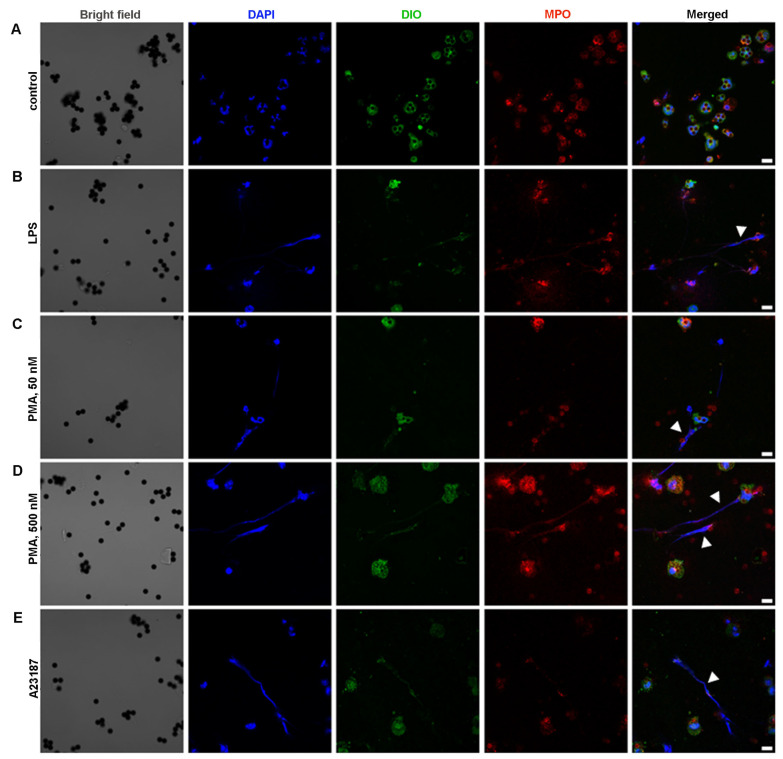
NET formation by spleen-derived neutrophils, isolated by IPS protocol. (**A**) Unstimulated neutrophils. (**B**) Neutrophils stimulated with 100 μg/mL LPS. (**C**,**D**). Neutrophils stimulated with 50 nM and 500 nM PMA, respectively. (**E**) Neutrophils stimulated with 5 μM A23187. Neutrophils were labeled with DAPI (blue signal, nuclei), DIOC6 (green signal, membranes), and anti-MPO (Red signal). Bright field images show the position of magnetic beads. Arrowheads indicate NETs. Scale bar, 10 µm. Images were obtained by confocal fluorescent microscopy.

**Figure 6 ijms-24-17273-f006:**
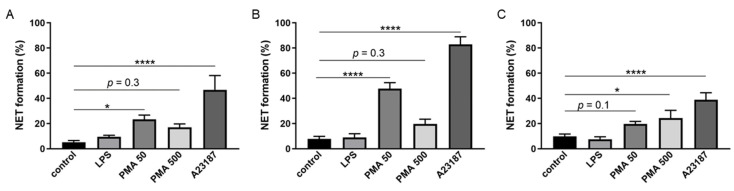
The percentage of neutrophils forming NETs in response to different stimulators. (**A**) BM-derived neutrophils isolated using the 2FLG protocol. (**B**) BM-derived neutrophils isolated using the INS protocol. (**C**) Splenic neutrophils isolated on Dynabeads. The isolated cells were stimulated with 100 μg/mL LPS, 50 or 500 nM PMA, and 5 μM A23187 for 3 h. Results are presented as mean ± SEM. Tukey’s multiple comparisons test was used. * *p* < 0.05, **** *p* < 0.0001.

**Figure 7 ijms-24-17273-f007:**
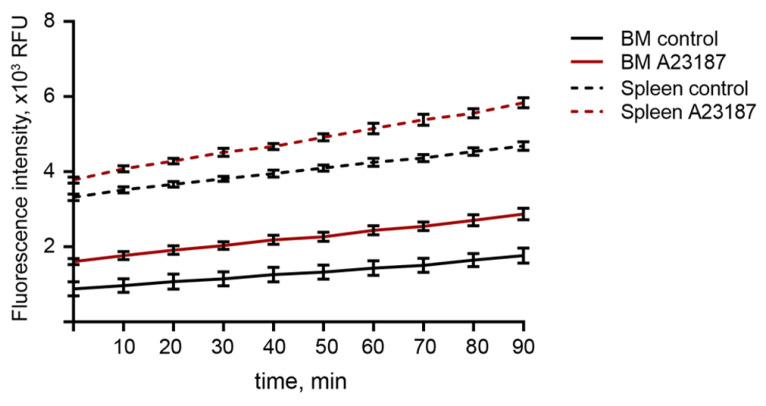
ROS production in murine neutrophils in response to calcium ionophore A23187. Mouse neutrophils isolated from bone marrow or spleens were preloaded with 10 μM 2′,7′-dichlorofluorescein diacetate (DCFDA) and either unstimulated (control) or activated with 5 μM A23187. The kinetics of ROS production was determined using a plate reader for 90 min with 10 min intervals. Values normalized by background subtraction (PBS) are shown. Results are presented as mean ± SD.

**Table 1 ijms-24-17273-t001:** Characteristics of cell samples isolated from the bone marrow of healthy C57BL/6 mice using 2FLG, 3FLG, INS and IPS protocols.

Method of Isolation	Sample Purity, %	Viability, %	General Yield, ×10^6^	Neutrophil Yield, ×10^6^	Cost
2FLG	72.4 ± 1.7	89.8 ± 1.4	8.6 ± 2.8	6.3 ± 2.1	the lowest
3FLG	76.7 ± 5.0	91.0 ± 2.9	0.5 ± 0.1	0.4 ± 0.04	middle
INS	80.3 ± 0.3	94.3 ± 0.8	10.7 ± 1.1	8.6 ± 0.9	the highest
IPS	99.3 ± 0.3	91.6 ± 0.3	4.3 ± 1.4	4.3 ± 1.4	high

**Table 2 ijms-24-17273-t002:** Characteristics of cell samples from spleen of healthy C57BL/6 mice isolated by2FLG, 3FLG, INS and IPS protocols.

Method of Isolation	Sample Purity, %	Viability, %	General Yield, ×10^6^	Neutrophil Yield, ×10^6^	Cost
2FLG	8.6 ± 1.2	74.0 ± 7.0	18.1 ± 1.9	1.5 ± 0.3	the lowest
3FLG	10.5 ± 1.7	86.0 ± 4.0	0.8 ± 0.1	0.08 ± 0.02	middle
INS	48.6 ± 4.3	89.6 ± 2.0	6.5 ± 2.4	3.3 ± 1.5	the highest
IPS	98.7 ± 0.5	88.5 ± 4.5	0.7 ± 0.25	0.69 ± 0.25	high

## Data Availability

The data are available from the corresponding author upon an email request.

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
