# Peer review of "A Comparative Study of Different Protocols for Isolation of Murine Neutrophils from Bone Marrow and Spleen"

_ijms, 2023, doi:10.3390/ijms242417273_

Round 1
Reviewer 1 Report
Comments and Suggestions for Authors
Abstract Line 14-15: I find it hard to agree with the assertion that obtaining neutrophils from biological materials such as bone marrow, blood, bronchoalveolar lavage, or spleen is difficult due to low concentration or small size. The provided protocol comparisons do not support this claim either. Therefore, this sentence might be omitted.
Figure 1: I suggest reclassifying this entire figure as a supplemental figure since it does not introduce new information, uncover previously unknown details, or contribute significantly to pre-existing knowledge. To enhance the figure's value, it would be beneficial for the author to include the total yield of each method, enabling a more comprehensive comparison of their efficiencies. Additionally, Table 1 might be better suited for presentation as Figure 1.
LDG Terminology: The use of "LDG" as an abbreviation for "Low Density Granulocyte" may lead to confusion. It is advisable to find an alternative designation to prevent potential misinterpretation.
Figure 2: I would appreciate further clarification on why neutrophils are gated as Ly6C negative, considering that neutrophils typically express about 10-100 times more Ly6C than lymphoid cells. Furthermore, it is essential to provide a clear rationale for the different gating strategies employed for the various isolation types in Figures 2 and 3.
IPS Data: I noticed the absence of IPS data in Table 1. Kindly ensure that this data is included to provide a comprehensive overview of the results.
Observations on Isolation Purity: Based on my experience across four labs over the past ten years, I have consistently observed purity levels exceeding 80% with the 2LDG protocol and over 90% with the Negative Selection protocol. Additionally, the EasySep™ Mouse Neutrophil Enrichment Kit claims 88% purity with mouse bone marrow cells. I am curious to understand the reasons behind the observed drop in isolation purity of almost 55% in the hands of the author.
Functional Tests and NETosis Evaluation: It is imperative to conduct functional tests, such as NADPH activation, Calcium flux, Degranulation, Phagocytosis, or NETosis, to assess the activation status of the cells following different isolation protocols. These tests would provide critical information that is currently missing from the manuscript.
Assessment of NETosis: The evaluation of NETosis solely based on the release of DNA may lead to misinterpretation, as extracellular DNA release can result from various cellular activities or cell death. To ensure accurate evaluation, the author should consider using a NET-specific marker and perform a quantitative assessment with H3Cit or MPO-DNA complex ELISA or immunofluorescence assays to confirm NETosis.
I hope these revised comments better convey your thoughts and suggestions to the author.
Comments on the Quality of English LanguageAcceptable
Author Response
Reviewer 1
Dear Reviewer,
We appreciate the time and effort that you have dedicated to revising our manuscript. Thank you for your valuable comments. We have revised the manuscript according to your comments and have attempted to cover all of them in the revised version. All text modifications are made in the editing mode using the “track changes” function. We provide a point-by-point answer to your comments:
1-Abstract Line 14-15: I find it hard to agree with the assertion that obtaining neutrophils from biological materials such as bone marrow, blood, bronchoalveolar lavage, or spleen is difficult due to low concentration or small size. The provided protocol comparisons do not support this claim either. Therefore, this sentence might be omitted.
Response: The sentence was deleted.
2-Figure 1: I suggest reclassifying this entire figure as a supplemental figure since it does not introduce new information, uncover previously unknown details, or contribute significantly to pre-existing knowledge. To enhance the figure's value, it would be beneficial for the author to include the total yield of each method, enabling a more comprehensive comparison of their efficiencies. Additionally, Table 1 might be better suited for presentation as Figure 1.
Response: We respectfully prefer not to reclassify Figure 1 as a supplemental figure. Since different protocols of isolation are used in this article we used this Figure just to illustrate these different protocols for better understanding.
3-LDG Terminology: The use of "LDG" as an abbreviation for "Low Density Granulocyte" may lead to confusion. It is advisable to find an alternative designation to prevent potential misinterpretation.
Response: The abbreviation 2LDG “layer density gradient” was changed to 2FLG “2 Ficoll layers density gradient”.
4-Figure 2: I would appreciate further clarification on why neutrophils are gated as Ly6C negative, considering that neutrophils typically express about 10-100 times more Ly6C than lymphoid cells. Furthermore, it is essential to provide a clear rationale for the different gating strategies employed for the various isolation types in Figures 2 and 3.
Response: In our gating strategy, Ly6C was used to identify monocyte population. However, since some neutrophil populations were shown to express Ly6C the gating strategy was changed, and neutrophils were identified as CD11b Ly6G double positive without considering Ly6C. The new gating strategy is simpler and shows higher isolation purity. The different gating strategies were removed and replaced with standardized one for all samples except the samples isolated using IPS, since the size of the events and granularity differ because of their binding with the magnetic beads. And these samples also were not stained with CD11b since after isolation, the magnetic beads could catch the CD11b antibodies and show false positive events.
5-IPS Data: I noticed the absence of IPS data in Table 1. Kindly ensure that this data is included to provide a comprehensive overview of the results.
Response: Bone marrow derived neutrophils were isolated using IPS method and the data was added.
6-Observations on Isolation Purity: Based on my experience across four labs over the past ten years, I have consistently observed purity levels exceeding 80% with the 2LDG protocol and over 90% with the Negative Selection protocol. Additionally, the EasySep™ Mouse Neutrophil Enrichment Kit claims 88% purity with mouse bone marrow cells. I am curious to understand the reasons behind the observed drop in isolation purity of almost 55% in the hands of the author.
Response: We agree that the isolation purity was lower than the isolation purity claimed by different authors in literature. However, one mistake was the gating strategy, after modifying the gating strategy the isolation purity was significantly higher. It is about 73 for 2LDG and 80 for EasySep kit.
7-Functional Tests and NETosis Evaluation: It is imperative to conduct functional tests, such as NADPH activation, Calcium flux, Degranulation, Phagocytosis, or NETosis, to assess the activation status of the cells following different isolation protocols. These tests would provide critical information that is currently missing from the manuscript.
Response: We agree with this suggestion. Along with NETosis, we performed the neutrophil oxidative burst test, to assess the activation status and functionality of isolated neutrophils. However, this point was added as a limitation in our study in the discussion section as follow:
“One limitation of this study is that the functional analysis included two functional tests of neutrophils: NETosis and ROS production analysis; however, these tests are one of multiple tests used to study neutrophil function. These tests include phagocytosis, microbe killing, degranulation assays, and chemotaxis. Blanter et al. [10] compared the functionality of human neutrophils, isolated on Ficoll or by immunomagnetic separa-tion, using different tests and found that immunomagnetic separation of neutrophils is more suitable for studying neutrophil polarization, phagocytosis, ROS production, degranulation, and NETosis, whereas density gradient purification is preferred for Boyden chemotaxis assays. Depending on that, we can conclude that in the case of NETosis or oxidative burst analysis, the protocols recommended in this study are of choice. However, when analyzing other neutrophil functions, immunomagnetic methods of negative selection may be more suitable.”
8-Assessment of NETosis: The evaluation of NETosis solely based on the release of DNA may lead to misinterpretation, as extracellular DNA release can result from various cellular activities or cell death. To ensure accurate evaluation, the author should consider using a NET-specific marker and perform a quantitative assessment with H3Cit or MPO-DNA complex ELISA or immunofluorescence assays to confirm NETosis.
Response: We stained the microscopic samples with MPO along with DAPI and DIO. NETs were confirmed to be stained with DAPI and MPO which is specific marker of NETs.
For quantification, we were unable to perform a quantitative analysis using ELISA or immunofluorescence assays; however, the percentage of neutrophils producing NETs in 5-nonoverlapping 20x fields was calculated to give our results more reliability.

Reviewer 2 Report
Comments and Suggestions for Authors
Dr. Khetam Sounbuli and colleagues compared the isolation of neutrophils from mouse bone marrow and spleen using different methods. They also compared the NET production capacity of the isolated neutrophils using each method. While there have been many publications using isolated neutrophils, few have compared these methods. In this regard, this paper is valuable in presenting the optimal methods for isolating neutrophils from bone marrow and spleen. However, since the focus of the paper is primarily methodological, the accuracy and reproducibility of the data are of utmost importance. The following improvements are needed:
<Major Remarks>
Lines 85-86: The authors state, "Various solutions - Ficoll, Percoll, Histopaque, Optiprep, Lympholyte – were used in density-gradient centrifugation protocols.", and then conclude that Ficoll is the best choice based on factors such as cell purity, yield, viability, and cost. I agree with the cost. However, when discussing cell purity, yield, and viability, it is important to provide data to support these claims. For example, the authors should provide reproducible experimental data as supplementary information showing that purity and yield did not differ significantly between these reagents.
Line 94: Please clarify in which solutions EDTA was added to minimize cell aggregation.
Figure 2, Figure 3: The scale on the x-axis in Fig. 2C and the scale on both the x-axis and y-axis in Fig. 3D differ from other figures in the same set, causing the neutrophil cell populations to appear displaced. Consistency of the scale and legends is crucial in presenting FACS data. Please make corrections.
Table 1, Table 2: Please define what "General Yield" means in the tables.
Line 167: The authors state, " general yield and neutrophil yield increase in order 3LDG<<2LDG<INS, and the more sharp difference (more than 17 fold) in general/neutrophil yield was between 3LDG and 2LDG protocols showing the applicability of 3LDG protocol for neutrophil isolation from bone marrow." Please clarify why the data suggest that the 3LDG protocol is applicable. It seems to suggest that 2LDG or INS protocols may be more desirable.
Lines 193-203: In the Results section, the order of reference to Figures 4C, 4D, and 4B is unnatural for a scientific paper. Additionally, Figure 4A should be briefly mentioned in the main text.
Lines 211-215: The text mentions 50 nM PMA for Figure 5B, but in Figure 5 it is labeled LPS. Such discrepancies may undermine the reliability of the data. Please make the necessary corrections.
Figure 4, Figure 5: Please consider providing quantitative data alongside the microscopic images, such as the percentage of neutrophils producing NETs in multiple fields of view.
Discussion:
Explain why NETs are not induced in splenic neutrophils with 500 nM PMA.
Provide reasonable discussion on why negative bead selection may not be suitable for isolation of splenic neutrophils.
<Minor Remarks>
If there are data on the IPS method tried in bone marrow, please present them.
On Line 91, it might be better to say, "As mentioned in the method section," since the Methods section likely appears further down in the final paper.
Author Response
Reviewer 2
Dear Reviewer,
We appreciate the time and effort that you have dedicated to revising our manuscript and thank you for your comments and suggestions, which have improved the manuscript.
The manuscript has been revised according to your comments. All text modifications are made in the editing mode using the “track changes” function. We provide a point-by-point answer to your comments:
1-Lines 85-86: The authors state, "Various solutions - Ficoll, Percoll, Histopaque, Optiprep, Lympholyte – were used in density-gradient centrifugation protocols.", and then conclude that Ficoll is the best choice based on factors such as cell purity, yield, viability, and cost. I agree with the cost. However, when discussing cell purity, yield, and viability, it is important to provide data to support these claims. For example, the authors should provide reproducible experimental data as supplementary information showing that purity and yield did not differ significantly between these reagents.
Response: This sentence was modified to “Various solutions - Ficoll, Percoll, Histopaque, Optiprep, Lympholyte – could be used in density-gradient centrifugation protocols. We selected Ficoll as the simplest, most accessible and the most widely used medium for density-gradient centrifugation (23).”
2-Line 94: Please clarify in which solutions EDTA was added to minimize cell aggregation.
Response: this sentence was deleted since different solutions in the different protocols contain EDTA and this sentence may lead to confusion.
11-Figure 2, Figure 3: The scale on the x-axis in Fig. 2C and the scale on both the x-axis and y-axis in Fig. 3D differ from other figures in the same set, causing the neutrophil cell populations to appear displaced. Consistency of the scale and legends is crucial in presenting FACS data. Please make corrections.
Response: We made correction and standardized the axis for all figures in flow cytometry plots, except the y-axis for neutrophils isolated on Dyanbeads (figure 2D, figure 3 D) since the SSC axis significantly differ from other samples due to the fact that neutrophils are bounded to the magnetic beads. Moreover, the gating strategy was simplified and neutrophils were defined as CD11b+ Ly6G+ without studying Ly6C.
12-Table 1, Table 2: Please define what "General Yield" means in the tables.
Response: the "General Yield" was defined in the revised manuscript in page as “is the cell yield obtained after neutrophil isolation without considering sample purity”
13-Line 167: The authors state, " general yield and neutrophil yield increase in order 3LDG<<2LDG<INS, and the more sharp difference (more than 17 fold) in general/neutrophil yield was between 3LDG and 2LDG protocols showing the applicability of 3LDG protocol for neutrophil isolation from bone marrow." Please clarify why the data suggest that the 3LDG protocol is applicable. It seems to suggest that 2LDG or INS protocols may be more desirable.
Response: Thank you for pointing this out. In this statement there was a mistake, and we agree that 2LDG or INS are more suitable. We corrected this sentence. However, to avoid the confusion “LDG” abbreviation was replaced with FLG since LDG is widely known abbreviation for low density granulocytes.
14-Lines 193-203: In the Results section, the order of reference to Figures 4C, 4D, and 4B is unnatural for a scientific paper. Additionally, Figure 4A should be briefly mentioned in the main text.
Response: we corrected the order of mentioning the figures in the text.
15-Lines 211-215: The text mentions 50 nM PMA for Figure 5B, but in Figure 5 it is labeled LPS. Such discrepancies may undermine the reliability of the data. Please make the necessary corrections.
Response: We reviewed and corrected the mention of figures in the text.
16-Figure 4, Figure 5: Please consider providing quantitative data alongside the microscopic images, such as the percentage of neutrophils producing NETs in multiple fields of view.
Response: NET quantification data was added.
Discussion:
17-Explain why NETs are not induced in splenic neutrophils with 500 nM PMA.
Response: NET quantification showed that 500 nM PMA cause NETosis in splenic neutrophils, however, not effective as in the concentration 50 nM. This point was discussed in the discussion more clearly.
18-Provide reasonable discussion on why negative bead selection may not be suitable for isolation of splenic neutrophils.
Response: The point was added in the discussion.
<Minor Remarks>
19-If there are data on the IPS method tried in bone marrow, please present them.
Response: Bone marrow derived neutrophils were isolated using IPS method and the data was added.
20-On Line 91, it might be better to say, "As mentioned in the method section," since the Methods section likely appears further down in the final paper.
Response: the sentence was corrected according to your comment.

Round 2
Reviewer 1 Report
Comments and Suggestions for Authors
1. Both recent and past images fail to offer adequate proof of NETosis structures, as MPO staining does not exhibit colocalization with DAPI-positive structures resembling NETs. Consequently, the reliability of NETosis quantification is questionable. Author refused to perform any other functional assay citing limitation that limits the novelty of the current submission.
2. The author reanalyzed flow data using a different gate following concerns about the sample's low purity. I observed a shift in the gate between Figure 3B's primary gate compared to Figures 3A and 3B. This discrepancy raises concerns about accepting the data in its current form, especially considering these findings are pivotal to the paper. Unfortunately, the author must replicate these experiments, incorporating CD45 gating after the single cell gate.
Comments on the Quality of English Language
All good.
Author Response
Reviewer 1
Dear reviewer, thank you for your comments. We appreciate your input to our manuscript. We revised the manuscript according to your comments. Point-by-point answers to your comments are provided below. The page and line numbers mentioned in the responses are in the revised version. All changes in the manuscript are made in red font color.
- Both recent and past images fail to offer adequate proof of NETosis structures, as MPO staining does not exhibit colocalization with DAPI-positive structures resembling NETs. Consequently, the reliability of NETosis quantification is questionable. Author refused to perform any other functional assay citing limitation that limits the novelty of the current submission.
Response:
According to the MPO staining, actually MPO exhibits colocalization with DAPI-positive structures, however, to a better visualization the contrast of the images was adjusted in ZEN software 2012 (Zeiss, Munich, Germany) to strength the signal of MPO. Please see figures 4 and 5 in the new version. The MPO signal is acceptable and the obtained images are similar to murine NETs obtained by other researchers.
We are confused with your comment that we refused to perform any other functional assay, because we did not refuse to perform other functional assays, we did perform neutrophil oxidative burst assay and the result of the assay were added to the manuscript in the past version and in this version (please see the section 2.5 ROS production in BM-derived neutrophils isolated using the 2FLG protocol and splenic neutrophils isolated using Dynabeads in response to stimulation in page 10 line 240 ). The results showed ROS production in neutrophils after activation with A23187. This assay is a functional assay and complement the results received from NETosis analysis, which why we chose this assay.
We know that there are many different tests to use to access neutrophil functionality, however, we choose to apply NETosis and neutrophil oxidative burst test to isolated neutrophils and this did not limit the novelty of our submission since few articles focus on diverse neutrophil isolation methods especially for murine, not human, neutrophils and particularly from murine spleen, the cells of which are not fully studied and few articles (cited in our manuscript) focus on their isolation.
- The author reanalyzed flow data using a different gate following concerns about the sample's low purity. I observed a shift in the gate between Figure 3B's primary gate compared to Figures 3A and 3B. This discrepancy raises concerns about accepting the data in its current form, especially considering these findings are pivotal to the paper. Unfortunately, the author must replicate these experiments, incorporating CD45 gating after the single cell gate.
Response:
First, we did not reanalyze flow data because of the sample’s purity, the data were reanalyzed in response to your valuable comment in the round 1 revision: “4-Figure 2: I would appreciate further clarification on why neutrophils are gated as Ly6C negative, considering that neutrophils typically express about 10-100 times more Ly6C than lymphoid cells. Furthermore, it is essential to provide a clear rationale for the different gating strategies employed for the various isolation types in Figures 2 and 3.”
Second, regarding to the shift in the Figure 3B, the different gates were applied in accordance with different guidelines on processing flowcytometry data which indicate to use the same gates in one sample batch. However, since, many factors could affect the location of cell population in the debris exclusion gate (gate 1), cell processing and isolation method could affect gate 1. Moreover, since homogeneous cell population will display a continuous distribution in dot plots, there was a shift in gate 1 in spleen and bone marrow neutrophils isolated using 3 FLG. However, to avoid this situation the gating strategy was standardized for all samples despite the isolation protocol. The gate 1 excludes debris, remains of dead cells or erythroblasts. The expression of CD45 in gate 1 was analyzed and the results show that gate 1 used in this study contain about 90% CD45 positive events (please see the figure below). The results confirm the applicability of our gating strategy (please, see attached file with figure).

Reviewer 2 Report
Comments and Suggestions for Authors
The authors adequately addressed my points. This paper will be a good atlas for neutrophil researchers. I appreciate the opportunity to review this study and agree to its publication.
Author Response
Dear reviewer, thank you very much for your careful study of our work and your opinion.
Round 3
Reviewer 1 Report
Comments and Suggestions for Authors
Please use the same magnification and scale bar to maintain consistency in Figure 4.